# Perinatal Use of Citrulline Rescues Hypertension in Adult Male Offspring Born to Pregnant Uremic Rats

**DOI:** 10.3390/ijms25031612

**Published:** 2024-01-28

**Authors:** You-Lin Tain, Chih-Yao Hou, Guo-Ping Chang-Chien, Sufan Lin, Chien-Ning Hsu

**Affiliations:** 1Department of Pediatrics, Kaohsiung Chang Gung Memorial Hospital, Kaohsiung 833, Taiwan; tainyl@cgmh.org.tw; 2Institute for Translational Research in Biomedicine, Kaohsiung Chang Gung Memorial Hospital, Kaohsiung 833, Taiwan; 3College of Medicine, Chang Gung University, Taoyuan 330, Taiwan; 4Department of Seafood Science, National Kaohsiung University of Science and Technology, Kaohsiung 811, Taiwan; chihyaohou@webmail.nkmu.edu.tw; 5Institute of Environmental Toxin and Emerging-Contaminant, Cheng Shiu University, Kaohsiung 833, Taiwan; guoping@csu.edu.tw (G.-P.C.-C.); linsufan2003@gmail.com (S.L.); 6Center for Environmental Toxin and Emerging-Contaminant Research, Cheng Shiu University, Kaohsiung 833, Taiwan; 7Super Micro Mass Research and Technology Center, Cheng Shiu University, Kaohsiung 833, Taiwan; 8School of Pharmacy, Kaohsiung Medical University, Kaohsiung 807, Taiwan; 9Department of Pharmacy, Kaohsiung Chang Gung Memorial Hospital, Kaohsiung 833, Taiwan

**Keywords:** chronic kidney disease, citrulline, nitric oxide, developmental origins of health and disease (DOHaD), asymmetric dimethylarginine, gut microbiota, hypertension

## Abstract

The growing recognition of the association between maternal chronic kidney disease (CKD) and fetal programming highlights the increased vulnerability of hypertension in offspring. Potential mechanisms involve oxidative stress, dysbiosis in gut microbiota, and activation of the renin–angiotensin system (RAS). Our prior investigation showed that the administration of adenine to pregnant rats resulted in the development of CKD, ultimately causing hypertension in their adult offspring. Citrulline, known for enhancing nitric oxide (NO) production and possessing antioxidant and antihypertensive properties, was explored for its potential to reverse high blood pressure (BP) in offspring born to CKD dams. Male rat offspring, both from normal and adenine-induced CKD models, were randomly assigned to four groups (8 animals each): (1) control, (2) CKD, (3) citrulline-treated control rats, and (4) citrulline-treated CKD rats. Citrulline supplementation successfully reversed elevated BP in male progeny born to uremic mothers. The protective effects of perinatal citrulline supplementation were linked to an enhanced NO pathway, decreased expression of renal (pro)renin receptor, and changes in gut microbiota composition. Citrulline supplementation led to a reduction in the abundance of *Monoglobus* and *Streptococcus* genera and an increase in *Agothobacterium Butyriciproducens*. Citrulline’s ability to influence taxa associated with hypertension may be linked to its protective effects against maternal CKD-induced offspring hypertension. In conclusion, perinatal citrulline treatment increased NO availability and mitigated elevated BP in rat offspring from uremic mother rats.

## 1. Introduction

Hypertension affects one in three adults worldwide, and its roots may begin in early life [1,2]. Identifying and addressing hypertension early on could be a cost-effective strategy to globally reduce its burden. The connection between early-life environmental influences and later-life diseases is known as the “developmental origins of health and disease (DOHaD)” [3]. Adverse maternal conditions during gestation can impact fetal programming, potentially resulting in offspring hypertension [4]. Previous research demonstrated that pregnant rats fed adenine exhibited reduced kidney function along with glomerular and tubulointerstitial damage, hypertension, and increased uremic toxins [5]. These characteristics closely mirror the complex nature of human CKD. Maternal adenine-induced CKD affects fetal programming, leading to offspring hypertension. This hypertension is associated with deficient nitric oxide (NO) signaling, abnormal activation of the renin–angiotensin system (RAS), and alterations in the composition of gut microbiota [5]. 

Dietary antioxidants play a significant role in the treatment and prevention of various human diseases [6]. Citrulline, a non-essential amino acid initially identified in watermelon, has emerged as a potential antioxidant supplement that contributes to the improvement of cardiometabolic health [7,8]. Evidence suggests that citrulline derived from watermelon serves as an antioxidant by supporting the synthesis of NO along with arginine [9]. Orally ingested citrulline is absorbed by enterocytes and efficiently transported to the kidneys, where it is transformed to arginine. Citrulline supplementation offers pharmacokinetic advantages over arginine by bypassing hepatic first-pass metabolism, leading to increased NO production [10]. As NO is a well-known vasodilator, oral supplementation of arginine or citrulline is believed to effectively reduce blood pressure (BP) [11].

Evidence suggests that early-life oxidative stress increases the future risk of hypertension [12]. Conversely, the perinatal use of dietary antioxidants has been shown to protect adult offspring from hypertension in various animal models [13]. Previously, we found that supplementing citrulline during pregnancy in rats with NO deficiency improved offspring hypertension by enhancing NO production [14]. Nevertheless, the impact of perinatal citrulline supplementation on offspring hypertension programmed by maternal uremia is still unknown.

The objective of this study was to investigate the protective role of citrulline in hypertension with developmental origins. In vivo, this was accomplished by administering perinatal citrulline supplementation in a maternal uremia rat model to delve into potential underlying mechanisms, including NO, the RAS, and gut microbiota.

## 2. Results

### 2.1. Body Weight and BP

Figure 1 depicts that at 12 weeks of age, there were no discernible differences in body weight (Figure 1A) and the kidney weight-to-body weight ratio (Figure 1B) among the four groups. Additionally, the plasma creatinine levels, serving as an indicator of kidney function in the CKD group, were comparable to those in the remaining groups (Figure 1C). BP measurements were conducted longitudinally from weeks 3 to 12 (Figure 1D). Notably, during weeks 8–12, maternal CKD resulted in an increase in offspring’s systolic BP, which was reversed by perinatal citrulline treatment (Figure 1D).

### 2.2. NO Pathway

Table 1 presents the findings regarding NO parameters in the plasma, including citrulline, arginine, asymmetric dimethylarginine (ADMA), and symmetric dimethylarginine (SDMA). Following perinatal citrulline supplementation, the citrulline concentration exhibited a significant increase in the NC group as opposed to the CKD and CKDC groups. Plasma concentrations of arginine were notably higher in the NC and CKDC groups when compared to the CKD group. The levels of ADMA and SDMA were elevated due to maternal CKD, and citrulline supplementation mitigated the increase in ADMA in the CKDC group. A noteworthy reduction in the arginine-to-ADMA ratio was observed in the CKD group, which was a trend that was counteracted by citrulline treatment (Table 1).

We next analyzed protein levels of dimethylarginine dimethylaminohydrolase-1 and -2 (DDAH1 and DDAH2; ADMA-metabolizing enzymes), endothelial NOS (eNOS), and neuronal NOS (nNOS) by Western blot. Their expression in the offspring’s kidneys is illustrated in Figure 2. Maternal rats with CKD led to a decrease in renal protein levels of eNOS and nNOS, which is an effect that was prevented by maternal citrulline treatment (Figure 2B,C). Figure 2D,E show that the renal expression of ADMA-metabolizing enzymes DDAH-1 and -2 was comparable across the four experimental groups.

The findings suggest that maternal CKD hinders the NO pathway by reducing eNOS and nNOS protein levels, the ratio of arginine to ADMA, and by elevating ADMA and SDMA concentrations. Maternal citrulline supplementation appears to counteract these effects, restoring NO availability by increasing arginine, the ratio of arginine to ADMA, eNOS, and nNOS, and decreasing ADMA.

### 2.3. RAS

Quantitative real-time polymerase chain reaction (qPCR) was employed to analyze various components of the RAS system. The components assessed included (pro)renin receptor (PRR), renin, angiotensin-converting enzyme (ACE), angiotensinogen (AGT), and angiotensin II type 1 receptor (AT1R). Renal mRNA content of renin, PRR, AGT, ACE, and AT1R did not differ between the N and CKD groups (Figure 3). Among the four groups, CKDC rats exhibited the lowest renal expression of PRR. 

### 2.4. Gut Microbiota Composition

Alpha diversity, representing the species richness and evenness within each sample, was assessed using Pielou’s evenness and the Shannon index. Both alpha-diversity indices exhibited no notable differences among the four groups (Figure 4A,B).

The Partial Least Squares Discriminant Analysis (PLS-DA) revealed distinct clustering of gut samples in the N group compared to the other groups. This indicated differences in gut microbiota between the N group and CKD group (*p* = 0.001 by ANOSIM), between the N group and NC group (*p* = 0.02 by ANOSIM), and between the N group and CKDC group (*p* = 0.016 by ANOSIM) (Figure 4C). However, when comparing the CKD group with the CKDC group, the observed differences did not reach statistical significance (*p* = 0.069 by ANOSIM).

To further assess the distinctions in gut microflora among the four groups, linear discriminant analysis effect size (LEfSe) analysis was conducted (Figure 5). CKD offspring rats exhibited a noteworthy rise in the abundance of genera *Turicibacter*, *Alistipes*, and *Neglectibacter*. Citrulline treatment led to an increased level of genera *Murimonas*, *Faecalimonas*, *Sinanaerobacter*, and *Breznakia* in the NC group. Additionally, Figure 5 highlighted that the genus *Peptococcus* was overrepresented in the CKDC group. Among these, certain taxa were found to be correlated with hypertension, including *Turicibacter*, *Alistipes*, *Faecalimonas*, and *Peptococcus* [14,15].

To test further whether certain microorganisms are involved in the protective role of citrulline against maternal CKD-primed offspring hypertension, we examined different compositions and abundance between the CKD group and CKDC group. At the genus level, compared with the CKD group, the proportions of *Monoglobus* and *Streptococcus* were lower in the CKDC group (Figure 6A,B). Species-based comparison showed the abundance of *Agothobacterium Butyriciproducens* was amplified by maternal citrulline supplementation in the CKDC rats in comparison to the CKD rats (Figure 6C). 

## 3. Discussion

Early-life oxidative stress serves as a critical mechanism in the developmental programming of hypertension, and antioxidant therapy emerges as a potential preventive strategy [12,13]. Our study presents the first evidence that offspring hypertension induced by maternal CKD can be prevented through perinatal citrulline supplementation. Key findings include: (1) maternal CKD induces offspring hypertension, which is a condition prevented by perinatal citrulline treatment; (2) hypertension in offspring primed by maternal CKD is linked to an inhibited NO pathway characterized by reduced eNOS and nNOS protein levels, a diminished ratio of arginine to ADMA, and increased ADMA and SDMA concentrations; (3) maternal citrulline treatment safeguards adult offspring from hypertension by restoring NO, decreasing renal PRR expression, and influencing gut microbiota; (4) the protective action of citrulline aligns with a decreased abundance of the genera *Monoglobus* and *Streptococcus* and an increase in *Agathobacterium Butyriciproducens*.

Previous research suggests that citrulline, functioning as an antioxidant, modulates NO and prevents oxidative stress-induced cardiovascular disease [8,9,10]. Our study extends the application of citrulline during gestation and lactation to mitigate offspring hypertension associated with maternal CKD. In alignment with previous studies involving models of maternal NO deficiency [16] and prenatal dexamethasone exposure [17], our data suggests that maternal citrulline supplementation enhances NO availability, averting offspring hypertension.

Our study’s results demonstrate that maternal CKD diminishes eNOS and nNOS protein levels in offspring kidneys, reduces the ratio of arginine to ADMA, and increases ADMA and SDMA, thus limiting NO production. Maternal citrulline therapy effectively reverses the inhibitory effects on NOS protein abundance and restores the balance between arginine and ADMA to enhance the NO pathway. Considering the dysregulated ADMA/NO pathway as a mediator of oxidative stress in hypertension [13], the beneficial action of citrulline may be linked to its ability to improve NO availability.

An activated classic RAS axis is known to increase BP through increased oxidative stress [18]. While the imbalance between NO and RAS is closely linked to hypertension pathophysiology [19], little is known about whether citrulline treatment can modulate the RAS to control BP. Activating PRR promotes vasoconstriction [20], and maternal citrulline supplementation reduces PRR, favoring lower BP. Although not statistically significant, our results suggest that citrulline treatment tends to reduce most classic RAS components. Future studies may explore whether citrulline’s protective effect against offspring hypertension correlates with RAS blockade.

Another potential protective mechanism of citrulline against maternal uremia-programmed hypertension may be associated with alterations in gut microbiota. The redox status influences gut health [21], and dietary antioxidants may benefit health by modulating gut microbiota [22]. Citrulline has been demonstrated to contribute to the maintenance of both the integrity of the intestinal barrier and the balance of microbiota [23]. However, there are no reports on the impacts of citrulline on gut microbiota in hypertension.

The PLS-DA analysis in our study did not reveal distinct clustering patterns between the CKD and CKDC groups, suggesting that citrulline supplementation may have a limited role in shaping offspring’s gut microbiota compared to maternal CKD. Nevertheless, citrulline still contributes to the low relative abundance of *Monoglobus* and *Streptococcus* and a high proportion of *Agathobacterium Butyriciproducens* in the gut microbiota of the CKDC group. The genus *Monoglobus* has been associated with hypertension [24], and *Streptococcus* spp., opportunistic pathogenic taxa, are often found in hypertensive gut microbiomes [25]. Our findings indicate that the protective effects of citrulline against hypertension in the offspring of uremic dams may be associated with its capacity to impact taxa associated with hypertension. *Agathobaculum butyriciproducens*, a butyrate-producing probiotic, has shown beneficial effects on cognitive deficits and Alzheimer’s disease pathologies [26]. Butyrate, a short-chain fatty acid (SCFA), can regulate BP through the activation of its receptors [27]. Previously, we observed that butyrate supplementation throughout gestation and lactation prevented offspring hypertension programmed by maternal CKD [28]. Citrulline likely has the potential to enhance SCFA-producing probiotics and, consequently, reduce BP. A previous study identified *Peptococcus* as bacteria depleted in subjects with metabolic syndrome [29]. Based on our LEfSe analysis, citrulline supplementation, which enhances *Peptococcus* abundance, may be attributed to its beneficial action in preventing hypertension.

Several limitations of the present study need acknowledgment. Firstly, we did not analyze gut microbiota and derived metabolites in offspring rats at different developmental stages and their dams. Gut microbial alterations in adult offspring rats may be attributed to postnatal plasticity rather than primary programmed processes responding to early-life environmental cues. Secondly, while we understand that the mechanisms mentioned may not entirely cover the antioxidant actions of citrulline against maternal CKD-programmed hypertension, a comprehensive examination of the complete mechanisms involved would aid in the development of novel antioxidant preventive therapies. Finally, our data, although useful in demonstrating that citrulline treatment has beneficial effects on male rat offspring’s BP, is limited to experimentation in this model. The beneficial effects of citrulline supplementation were associated with the restoration of the NO pathway and modifications in gut microbiota, yet additional underlying mechanisms remain to be fully elucidated. While dietary antioxidants present a promising strategy for oxidative-stress-induced hypertension, conclusive results in humans are still pending [30,31]. Further investigations are required in other models of programmed hypertension and in humans before clinical translation.

## 4. Materials and Methods

### 4.1. Animals

We procured female Sprague–Dawley (SD) rats aged 6–8 weeks from BioLASCO Taiwan Co., Ltd. (Taipei, Taiwan) and housed them in our AAALAC-accredited animal facility. All procedures adhered to the regulations set by the Institutional Animal Care and Use Committee (IACUC) at our hospital with the permit number 2022091601.

Figure 7 illustrates the experimental protocol. Eight-week-old female rats (n = 12) were randomly divided into two groups. The rats were assigned to either a normal diet or a 0.5% adenine diet for a duration of 3 weeks, as previously described [5]. Individual females were paired overnight with a proven fertile male until the identification of a copulatory plug. Pregnant rats were then randomized into four groups: rats receiving a normal diet (N), adenine-treated rats (CKD), control rats receiving citrulline supplementation (0.1% citrulline in drinking water) throughout gestation and lactation (NC), and CKD rats receiving citrulline supplementation (CKDC). The citrulline dosage followed a previously established protocol [16]. Litter sizes at birth were reduced to eight pups. Given the higher prevalence of hypertension in males compared to females [32], only male offspring from each litter were selected for subsequent experiments.

Rat offspring were assigned to four groups (8 animals each): N, CKD, NC, and CKDC. To acclimate the rats, we utilized the CODA BP system (a tail-cuff method, Kent Scientific Corporation, Torrington, CT, USA) for BP measurements every four weeks. At 12 weeks of age, blood draws and sacrifices were performed to assess kidney weights. Kidney tissues were snap-frozen and stored accordingly. Prior to sacrifice, fecal samples from each offspring were stored in a −80 °C freezer. The creatinine concentrations in rat offspring blood were determined using high-performance liquid chromatography (HPLC, HP Agilent 1100, Agilent Technologies Inc., Santa Clara, CA, USA).

### 4.2. NO Parameters

The HPLC method was employed to analyze plasma levels of NO-related parameters. The measurements were made on an Agilent 1100 HPLC (Agilent Technologies Inc.) by using O-phthalaldehyde/3-mercaptopropionic acid (OPA/3-MPA) as a derivatization agent with fluorescence detection. The ratio of arginine to ADMA was calculated, which provides information on NO availability [33]. 

### 4.3. Western Blot 

Equal amounts of 200 µg of kidney cortical proteins were loaded per lane onto a polyacrylamide gel and subjected to electrophoresis. Following separation, the proteins were transferred onto nitrocellulose membranes. Membranes underwent treatment with a 0.1% Ponceau S solution (Sigma-Aldrich, St. Louis, MO, USA) for 10 min on a shaker, which was followed by rinsing with distilled water to eliminate background staining. Ponceau S staining served for total protein normalization.

The transferred proteins were then probed using specific antibodies, including a mouse eNOS antibody (1:250; BD610297BD, Biosciences, San Jose, CA, USA), a mouse nNOS antibody (1:200; SC-5302, Santa Cruz, CA, USA), a mouse DDAH1 antibody (1:500; SC-271337, Santa Cruz), or a rabbit DDAH2 antibody (1:2000; Ab184166, Abcam, Cambridge, UK). Subsequent to washing, the blots were incubated with the corresponding secondary antibody conjugated to horseradish peroxidase. Immunopositive bands were scanned using an imaging densitometer (Quantity One, Bio-Rad, Hercules, CA, USA) to quantify integrated optical density (IOD). Protein abundance was expressed as IOD normalized by Ponceau S stain (PonS, representing the total protein loaded). Complete blots and corresponding images of Ponceau S staining can be found in Appendix A.

### 4.4. Analysis of RAS Components Using qPCR

Total RNA was isolated from renal cortical tissues for qPCR analysis on a thermal cycler (iCycler, Bio-Rad, Hercules, CA, USA) in duplicate. The internal control utilized in this study was the 18S ribosomal RNA (R18S). PCR primers for both RAS components and R18S are detailed in Table 2. Relative quantification was determined through the comparative 2^−ΔΔCT^ method.

### 4.5. 16S rRNA Sequencing

Microbial community DNA was extracted from fecal samples and subsequently underwent 16S rRNA sequencing at Biotools Co., Ltd. (New Taipei City, Taiwan) [5]. The amplification of the V1–V9 region of the 16S rRNA gene with barcoded primers was prepared for a multiplexed SMRTbell library (PacBio, Menlo Park, CA, USA) and the sequencing procedure. To construct a phylogenetic tree, QIIME2 phylogeny fast tree utilized a set of sequences representing the amplicon sequence variants (ASVs) [34,35]. Alpha diversity analysis, evaluating microbiota richness and evenness within a single sample, utilized the Shannon index and Pielou’s evenness. Beta diversity analysis relied on ANOSIM and PLS-DA. Significantly differential taxa were identified using LEfSe analysis with an LDA score exceeding 3.

### 4.6. Statistics

The data are expressed as means ± standard error of the mean (SEM). Group distinctions were evaluated utilizing either one-way ANOVA or two-way ANOVA, depending on the context. Subsequent to the ANOVA, Tukey post hoc analysis was conducted to elucidate differences between specific groups. A significance level of *p*  <  0.05 was employed to determine statistical significance. All statistical analyses were carried out using SPSS 17.0 software (SPSS, Inc., Chicago, IL, USA).

## 5. Conclusions

This study represents one of the initial observations highlighting the potential of citrulline supplementation during gestation and lactation to prevent offspring hypertension complicated by maternal uremia. Given the reversible nature of offspring hypertension through citrulline, a deeper understanding of its extent and the involved mechanisms could contribute to the development of optimal antioxidants as preventive therapies, thereby mitigating the health burden imposed by elevated BP on future generations.

## Figures and Tables

**Figure 1 ijms-25-01612-f001:**
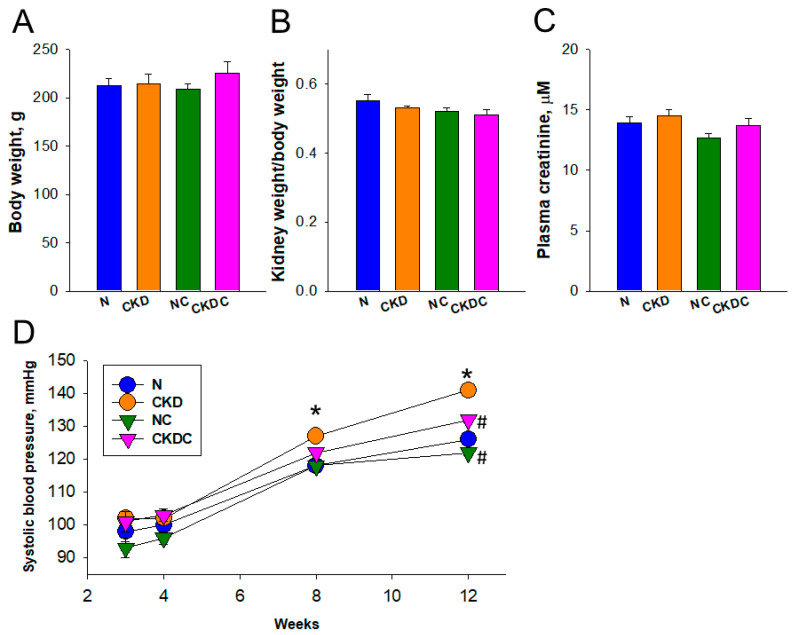
(**A**) Offspring body weight, (**B**) ratio of kidney weight-to-body weight, and (**C**) plasma creatinine concentration at 12 weeks of age. (**D**) Systolic blood pressure in offspring from 3 to 12 weeks of age with a sample size of n = 8 per group. * *p* < 0.05 vs. N; # *p* < 0.05 vs. CKD.

**Figure 2 ijms-25-01612-f002:**
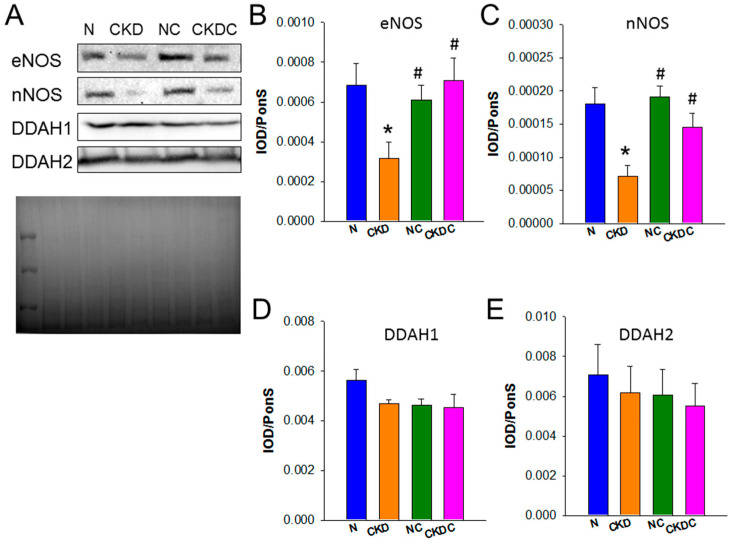
The results of Western blot analyses for (**A**) endothelial nitric oxide synthase (eNOS), neuronal NOS (nNOS), dimethylarginine dimethylaminohydrolase-1 (DDAH1), and -2 (DDAH2) in the offspring’s kidneys, with Ponceau S staining employed as a loading control. The relative abundance of (**B**) eNOS, (**C**) nNOS, (**D**) DDAH1, and (**E**) DDAH2 was quantified and presented. * *p* < 0.05 vs. N; # *p* < 0.05 vs. CKD.

**Figure 3 ijms-25-01612-f003:**
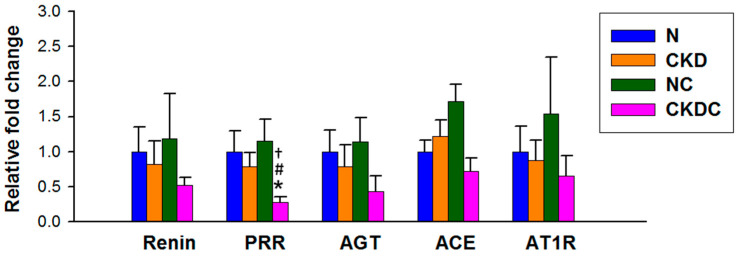
Renal mRNA expression of RAS components. * *p* < 0.05 vs. N; # *p* < 0.05 vs. CKD; † *p* < 0.05 vs. NC.

**Figure 4 ijms-25-01612-f004:**
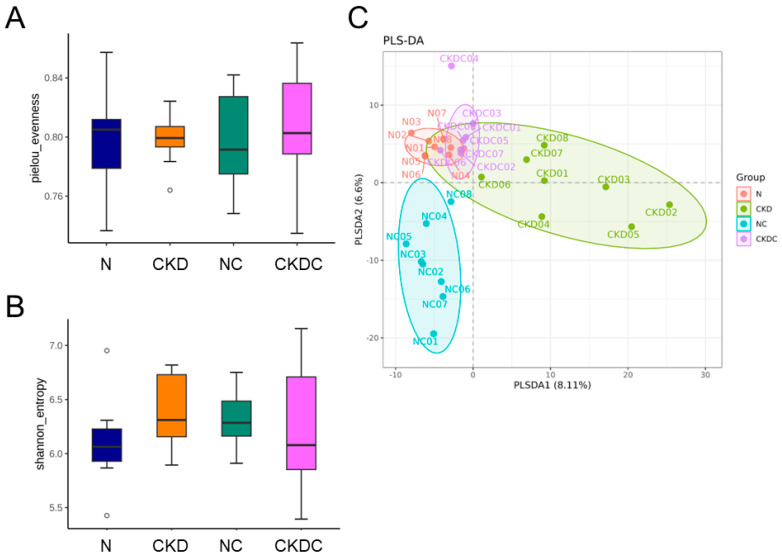
(**A**) Pielou’s evenness and (**B**) Shannon index, illustrating alpha diversity across the four groups. (**C**) The Partial Least Squares Discriminant Analysis (PLS-DA) plots depict the clustering of fecal microbiota from the four groups. Each dot represents an individual’s microbiota, with color indicating the respective group.

**Figure 5 ijms-25-01612-f005:**
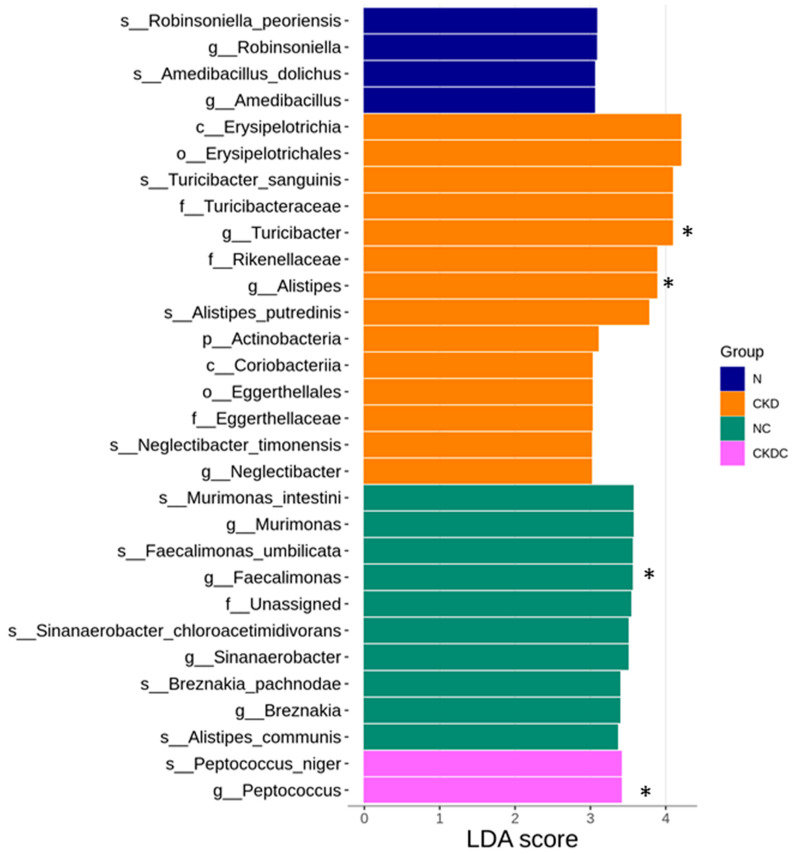
Linear discriminant analysis effect size (LEfSe) illustration. Different colors denote groups of microbes that have significant effects in different groups with linear discriminant analysis (LDA) > 3. * indicates taxa that are linked to hypertension.

**Figure 6 ijms-25-01612-f006:**
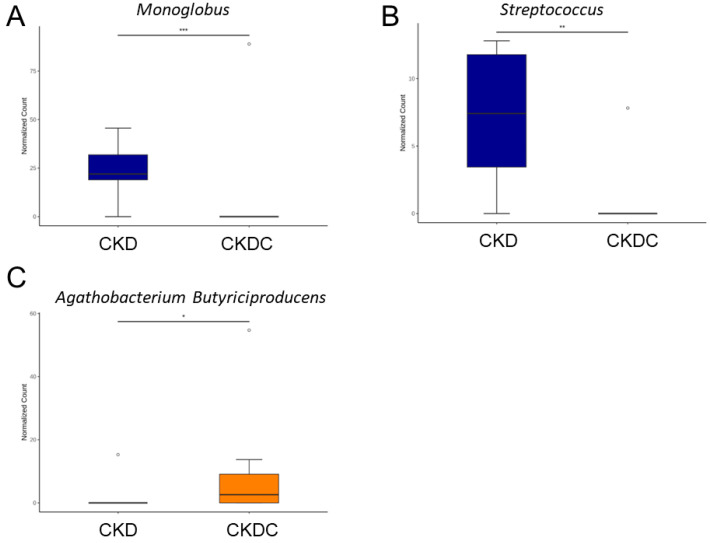
Relative abundance of (**A**) *Monoglobus* (genus), (**B**) *Streptococcus* (genus), and (**C**) *Agothobacterium Butyriciproducens* (species) between the CKD and CKDC group. * *p* < 0.05; ** *p* < 0.01; *** *p* < 0.005.

**Figure 7 ijms-25-01612-f007:**
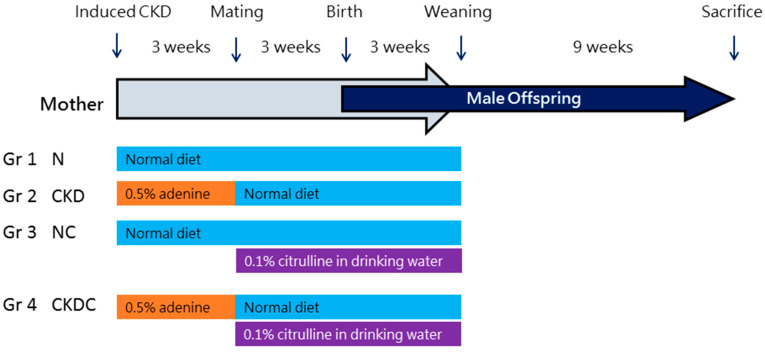
Experimental protocol used in the current study. N = dams received a normal diet; CKD = dams treated with an adenine diet; NC = dams received a normal diet and received citrulline supplementation; CKDC = adenine-treated dams received citrulline supplementation.

**Table 1 ijms-25-01612-t001:** NO parameters in the plasma of 12-week-old offspring.

Groups	N	CKD	NC	CKDC
Citrulline, μM	65.6 ± 3.1	63.4 ± 1.9	74.1 ± 3 #	63.3 ± 1.8 †
Arginine, μM	173.6 ± 15.3	154.3 ± 3.5	191.7 ± 7.6 #	172.4 ± 3.7 #†
ADMA, μM	2.04 ± 0.09	2.65 ± 0.09 *	1.81 ± 0.19 #	2.07 ± 0.04 #†
SDMA, μM	1.5 ± 0.08	2.16 ± 0.12 *	1.7 ± 0.16 #	1.91 ± 0.09
Ratio of arginine-to-ADMA	84.6 ± 5.3	57.7 ± 3.2 *	111.1 ± 7.5 *#	83.6 ± 2.9 #†

N = 8/group. * *p* < 0.05 vs. N; # *p* < 0.05 vs. CKD; † *p* < 0.05 vs. NC.

**Table 2 ijms-25-01612-t002:** Primers for qPCR.

Gene	Accession No	Sense	Antisense
AGT	XM_032887807.1	5 gcccaggtcgcgatgat 3	5 tgtacaagatgctgagtgaggcaa 3
Renin	J02941.1	5 aacattaccagggcaactttcact 3	5 acccccttcatggtgatctg 3
ACE	U03734.1	5 caccggcaaggtctgctt 3	5 cttggcatagtttcgtgaggaa 3
PRR	AB188298.1	5 gaggcagtgaccctcaacat 3	5 ccctcctcacacaacaaggt 3
AT1R	NM_030985.4	5 gctgggcaacgagtttgtct 3	5 cagtccttcagctggatcttca 3
R18S	X01117	5 gccgcggtaattccagctcca 3	5 cccgcccgctcccaagatc 3

## Data Availability

Data are contained within the article.

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
