# Peer review of "Perinatal Use of Citrulline Rescues Hypertension in Adult Male Offspring Born to Pregnant Uremic Rats"

_ijms, 2024, doi:10.3390/ijms25031612_

Round 1
Reviewer 1 Report
Comments and Suggestions for Authors
In this study, the Authors aimed to investigate of the preventive effects of perinatal citrulline supplementation offspring hypertension induced by maternal kidney disease. The key findings consist in the uncovering of long-term beneficial effects of cutrulline on blood pressure and microbiome of the experimental animals.
One of the strong sides of this study is that it uncovers the clinical and therapeutic importance of the perinatal period. It also suggests a new therapy based on citrulline supplementation.
This article can be published. There are just several small concerns:
Abstract:
1 Please, insert just two lines explaining the detrimental effects of adenine- containing diet into background.
2 Please, explain physiological significance of the microbiome shift resulted from citrulline treatment.
Introduction:
1 Please, explain in more details how exactly adenine diet causes long-term detrimental effects on kidney and why it triggers hypertension? In other words, please, describe the results of the Ref 5 more thoroughly.
2 In your study, postnatal citrulline treatment mitigated the detrimental effects of adenine-rich diet survived in pregnancy. It is a very important result that underscores the importance perinatal window. Currently, almost all studies are focused on the fetal programming of adult diseases, whereas perinatal period seems to escape notice. There are just several studies indicating that perinatal period is also important for developmental programming. For example, in accordance with your study, it was recently demonstrated that oxidative stress triggered by systemic inflammation causes long-term cardiac pathology (10.3390/ijms24087063 )
3. How do you think, how citrullin supplementation influences postnatal cell proliferation? Whether is it able to restore cell deficiency and atrophy originating from pregnant period?
Methods:
1 It would be good to provide an explanatory figure illustrating the scheme of the experiment where a Reader can see the age of the animals at all important points. This figure also can be included in a Graphical Abstract (please. Prepare if possible).
Results:
Please, specify at rhe Fig 5 or prepare a new figure indicating useful bacteria and detrimental ones.
Author Response
RESPONSES TO REVIEWER’S COMMENTS
Reviewer #1
In this study, the Authors aimed to investigate of the preventive effects of perinatal citrulline supplementation offspring hypertension induced by maternal kidney disease. The key findings consist in the uncovering of long-term beneficial effects of cutrulline on blood pressure and microbiome of the experimental animals.
One of the strong sides of this study is that it uncovers the clinical and therapeutic importance of the perinatal period. It also suggests a new therapy based on citrulline supplementation.
This article can be published. There are just several small concerns:
RESPONSE: We thank Reviewer #1 for his/her generous support.
Abstract:
1 Please, insert just two lines explaining the detrimental effects of adenine- containing diet into background.
RESPONSE: As per suggestion, we have incorporated the following sentence into the Abstract. “Our prior investigation showed that the administration of adenine to pregnant rats resulted in the development of CKD, ultimately causing hypertension in their adult offspring.”
2 Please, explain physiological significance of the microbiome shift resulted from citrulline treatment.
RESPONSE: We have incorporated the following sentence into the Abstract. “Citrulline's ability to influence taxa associated with hypertension may be linked to its protective effects against maternal CKD-induced offspring hypertension.”
Introduction:
1 Please, explain in more details how exactly adenine diet causes long-term detrimental effects on kidney and why it triggers hypertension? In other words, please, describe the results of the Ref 5 more thoroughly.
RESPONSE: We have rephrased our statement to provide a comprehensive description of this model.
“Previous research demonstrated that pregnant rats fed adenine exhibited reduced kidney function, along with glomerular and tubulointerstitial damage, hypertension, and increased uremic toxins [5]. These characteristics closely mirror the complex nature of human CKD. Maternal adenine-induced CKD affects fetal programming, leading to offspring hypertension. This hypertension associated with deficient nitric oxide (NO) signaling, abnormal activation of the renin-angiotensin system (RAS), and alterations in the composition of gut microbiota [5].”
2 In your study, postnatal citrulline treatment mitigated the detrimental effects of adenine-rich diet survived in pregnancy. It is a very important result that underscores the importance perinatal window. Currently, almost all studies are focused on the fetal programming of adult diseases, whereas perinatal period seems to escape notice. There are just several studies indicating that perinatal period is also important for developmental programming. For example, in accordance with your study, it was recently demonstrated that oxidative stress triggered by systemic inflammation causes long-term cardiac pathology (10.3390/ijms24087063 )
RESPONSE: Unlike humans, kidney development in rats lasts until 1–2 week after birth. Hence, in the current study, citrulline treatment was administered during gestation and lactation to encompass the entire period of nephrogenesis. Since we did not conduct a comparative analysis of citrulline treatment at various developmental stages (e.g., gestation only, lactation only, and both gestation and lactation), it is challenging to determine the most crucial developmental window. Nevertheless, we acknowledge the importance of this aspect and identify it as a topic worthy of further exploration in future studies.
- How do you think, how citrullin supplementation influences postnatal cell proliferation? Whether is it able to restore cell deficiency and atrophy originating from pregnant period?
RESPONSE: Regrettably, we are unable to address the Reviewer's inquiry here, as cell deficiency and atrophy do not represent predominant phenotypes in this particular model. Nevertheless, considering the involvement of NO in cellular adaptation and proliferation (Redox Biol. 2020 Jul;34:101550. doi: 10.1016/j.redox.2020.101550.), there is a potential that citrulline supplementation could enhance NO availability, thereby potentially ameliorating fetal programming-induced cell deficiency and atrophy. We acknowledge the significance of this matter and recognize it as a subject worthy of further investigation in future studies.
Methods:
1 It would be good to provide an explanatory figure illustrating the scheme of the experiment where a Reader can see the age of the animals at all important points. This figure also can be included in a Graphical Abstract (please. Prepare if possible).
RESPONSE: Following the suggestion, we have added a Figure 7 to illustrate the experimental protocol.
Results:
Please, specify at the Fig 5 or prepare a new figure indicating useful bacteria and detrimental ones.
RESPONSE: As recommended, we have highlighted in Figure 5 the particular taxa associated with hypertension.

Reviewer 2 Report
Comments and Suggestions for Authors
The manuscript entitled "Perinatal Use of Citrulline Rescues Hypertension in Adult Male Offspring Born to Pregnant Uremic Rats" presents the results of an experimental study performed in uremic rats, which was induced in female Sprague–Dawley rats by administering adenine in the diet. According to the results, the supplementation of citrulline in drinking water prevented the development of hypertension due to maternal uremia. To elucidate the effects induced by citrulline several components of the NO pathway were measured at mRNA and protein levels. On the other hand, the parameters describing chronic kidney disease were also monitored. Almost all the parameters showed that citrulline administration could prevent CKD induced modifications on NO pathway and RAS system.
Further evidence has been provided that CKD influenced the gut microbiota composition, and citrulline had a beneficial effect in the control group. However, this was not significant in the CKD group.
Overall, the manuscript needs minor modifications only.
First, the objective of the study should be more specific than "exploring the underlying protective mechanisms".
Second, the Conclusions section should not contain references. The main focus should be kept on the results of the current study. The additional information should be discussed elsewhere.
Third, high-resolution images of WB should be published as supplementary files.
Author Response
Reviewer #2
The manuscript entitled "Perinatal Use of Citrulline Rescues Hypertension in Adult Male Offspring Born to Pregnant Uremic Rats" presents the results of an experimental study performed in uremic rats, which was induced in female Sprague–Dawley rats by administering adenine in the diet. According to the results, the supplementation of citrulline in drinking water prevented the development of hypertension due to maternal uremia. To elucidate the effects induced by citrulline several components of the NO pathway were measured at mRNA and protein levels. On the other hand, the parameters describing chronic kidney disease were also monitored. Almost all the parameters showed that citrulline administration could prevent CKD induced modifications on NO pathway and RAS system.
Further evidence has been provided that CKD influenced the gut microbiota composition, and citrulline had a beneficial effect in the control group. However, this was not significant in the CKD group.
Overall, the manuscript needs minor modifications only.
RESPONSE: We thank the reviewer #2 for the efforts and the constructive comments on the work.
First, the objective of the study should be more specific than "exploring the underlying protective mechanisms".
RESPONSE: We have revised our statement to precisely articulate our objective.
“The objective of this study was to investigate the protective role of citrulline in hypertension with developmental origins. In vivo, this was accomplished by administering perinatal citrulline supplementation in a maternal uremia rat model to delve into potential underlying mechanisms, including NO, the RAS, and gut microbiota.”
Second, the Conclusions section should not contain references. The main focus should be kept on the results of the current study. The additional information should be discussed elsewhere.
RESPONSE: Following the suggestion, we have relocated the mentioned segment to the Discussion section.
“The beneficial effects of citrulline supplementation were associated with the restoration of the NO pathway and modifications in gut microbiota, yet additional underlying mechanisms remain to be fully elucidated. While dietary antioxidants present a promising strategy for oxidative-stress-induced hypertension, conclusive results in humans are still pending [30,31].”
Third, high-resolution images of WB should be published as supplementary files.
RESPONSE: Following the suggestion, we have added high-resolution images of WB as supplementary file S1.

Round 2
Reviewer 1 Report
Comments and Suggestions for Authors
The Authors addressed all my comments.
The Article can be published.